# Data-driven evaluation of the Boston marathon qualifying times

**Laura Albrecht** *, Ross Ring-Jarvi, Dorit Hammerling

Department of Applied Mathematics and Statistics, Colorado School of Mines, Golden, CO, United States of America

* lalbrecht@mines.edu

**Data Availability Statement:** All data was taken from a public website (www.marathonguide.com). The clean version of our data is uploaded to zenodo currently. doi: 10.5281/zenodo.6959864.

## Abstract

The Boston Marathon is one of the most prestigious running races in the world. From its inception in 1897, popularity grew to a point in 1970 where qualifying times were implemented to cap the number of participants. Currently, women's qualifying times in each age group are thirty minutes slower than the men's qualifying times equating to a 16.7% adjustment for the 18-34 age group, decreasing with age to a 10.4% adjustment for the 80+ age group. This setup somewhat counter-intuitively implies that women become faster with age relative to men. We present a data-driven approach to determine qualifying standards that lead to an equal proportion of qualifiers in each age category and gender. We had to exclude the 75-79 and 80+ age groups from analysis due to limited data. When minimizing the difference in proportion of men and women qualifying, the women's times for the 65-69 and 70-74 age groups are 4-5 minutes slower than the current qualifying standard, while they are 0 to 3 minutes faster for all other age groups.

## Introduction

Every year on Patriots Day thousands of runners line up to take part in the historic Boston Marathon. From its inception in 1897, popularity grew to a point in 1970 where qualifying times had to be implemented to cap the total number of participants [1]. Runners must achieve a certain qualifying time at a previous marathon to be eligible to register for the Boston Marathon. The increase in popularity of both the Boston Marathon and marathon running in general has caused the Boston Athletic Association (BAA) to make the time standards more stringent three times in the last twenty years. The most recent time standard drop of five minutes in each age group, implying faster qualifying times, went into effect in 2020. A runner's age on the date of the Boston Marathon and the gender they self-identified at the time of their qualifying race determines their required qualifying time [2]. We utilize this definition of gender in our analysis and only use the term sex in reference to previous work.

**Funding:** The authors received no specific funding for this work.

**Competing interests:** The authors have declared that no competing interests exist.

The new qualifying standards, given in Table 1, start at 3 hours for men and get slower on a sliding scale across each subsequent age group. The women's qualifying times (QTs) are 30 minutes slower than the men's times in every age group. This time gap of 30 minutes equates to a 16.7% adjustment for the 18–34 age group decreasing with age to a 10.4% adjustment for the 80+ age group implying women become faster with age relative to men.

The effects of sex or age on running performance have been the topic of much previous research. Performance for elite and average runners has been found to decline around age 35 and 50, respectively [3, 4]. The difference between the average male and female runner has been estimated to be about 11.6% [5]. However, not much research has assessed if this gap is consistent between men and women's times as they age. One study of the Berlin Marathon times speculated the decline in performance with aging may be different for men and women [6]. When the BAA was considering updating the qualifying times in 2013, Smith et al. [7] investigated the potential effect of changing the qualifying times by different amounts based on the age-sex distribution of runners that qualify. They observed a substantially lower percentage of female qualifiers compared to male qualifiers in the 60 and over age groups.

Women were first allowed to enter the Boston marathon in 1972 and at the time had to achieve the same qualifying time as the men. Small changes between the men's and women's times were introduced over the next 15 years and in 1987 the static 30-minute time difference between men and women was introduced. While the time standards themselves have changed multiple times since then, the 30-minute gap between men's and women's times in all age groups has remained consistent. The 70–74, 75–79, and 80+ age groups were added in 2002, again adopting the same 30-minute time gap that existed at the younger age groups. To the best of our knowledge, the decision to use this fixed time difference in every age group was not informed by data.

First, we investigate the effect of this 30-minute time gap and the distribution of runners at different age groups and genders. Then, we present one potential data-driven approach to suggest a more quantitatively justified method for setting future qualifying times. We aim to determine qualifying times leading to an equal proportion of qualifiers across genders. That is, if, for example, 7% of men 18–34 reach the qualifying time, we seek a qualifying time for women leading to 7% of women 18–34 qualifying as well. Seeking a fixed proportion of qualifiers across one race or after combining results from all races would be relatively simple. However, the proportion of qualifiers in each marathon and year are highly impacted by different features of that race. The proportion of runners qualifying in the same marathon in different years is influenced by factors such as weather and the individual runners participating in a given year. When comparing different marathons, other factors are also introduced such as elevation, course profile, and prize money. To control for this variability, we seek to find qualifying times that minimize the difference in the proportion of qualifiers conditioning on each individual marathon and year. Boston marathon qualifying times have historically been set at integer values in minutes, which we adopt. Therefore, our goal is to find integer qualifying times for women that minimize the difference in the proportion of qualifiers across all age groups, genders, marathons, and years using a large sample of popular Boston-qualifying marathons for a time period of 20 years.

**Table 1. 2020 Boston qualifying times.** Times are presented in H:MM format.

| Age Group | 18–34 | 35–39 | 40–44 | 45–49 | 50–54 | 55–59 | 60–64 | 65–69 | 70–74 | 75–79 | 80+ |
|---|---|---|---|---|---|---|---|---|---|---|---|
| Men's QT | 3:00 | 3:05 | 3:10 | 3:20 | 3:25 | 3:35 | 3:50 | 4:05 | 4:20 | 4:35 | 4:50 |
| Women's QT | 3:30 | 3:35 | 3:40 | 3:50 | 3:55 | 4:05 | 4:20 | 4:35 | 4:50 | 5:05 | 5:20 |

## Materials and methods

### Data

We developed a web-scraper in Python to collect results from www.marathonguide.com [8] for 10 different marathons from 2000–2019; Chicago, New York, Los Angeles, Twin Cities, Houston, Philadelphia, California International, Marine Corps, Grandma's, and Honolulu Marathon. We chose these marathons to get a good geographic sample of races across ten of the top largest marathons in the US [9]. Additionally, all marathons except for Honolulu are listed as top qualifying marathons or as one of the marathons with the most Boston qualifiers [8, 10]. We used the Selenium package [11] to cycle through all pages of results and then use the Beautiful Soup package [12] to change the HTML race content into a Python string object. R Tidyverse [13] was used to clean and organize the data. A few marathon observations are missing or excluded in our analysis; New York 2012 was canceled due to super-storm Sandy, Twin Cities in 2003 and 2005 as well as Grandma's in 2006 are excluded due to age being missing from the results. Additionally, the 75–79 and 80+ age groups have very few women runners and have been excluded from our analysis. This data is publicly available from Marathon Guide and our analysis complied with the websites terms of service [8].

Overall, our collected data set contains 3.2 million runners, 1.9 million men (59%) and 1.3 million women (41%). Fig 1 shows the number of runners and the percentage that qualify for Boston in each age group in our data. The majority of runners in our data are in the 18–34 age group. The proportion of women runners starts at 49.5% for the 18–34 age group and decreases to approximately 21% for the 70–74 age group. The percentage of runners that qualify for Boston in each age group ranges from about 4–9%. Other than the 18–34 age group, women qualify at a higher rate for all ages 50 and under. For age groups 55 and over, the percentage of men that qualify is greater and the gap between the proportion of men and women qualifying increases in each subsequent age group (Fig 1).

Our collected data are used to analyze the distribution of runners that qualify for Boston at the current qualifying times and compare those times with different potential qualifying standards. As an initial evaluation of the current 30-minute time gap between men's and women's qualifying times, we looked at the difference between men's and women's times at fixed percentiles in our data set. We determined the time in our data correlated with the top 4–9% of men and women, respectively, at each 0.1% increment. This covers the range of the percentage of runners that qualify in each age group at the current Boston Qualifying Times as seen in Fig 1.

### Minimizing the difference in the proportion of qualifiers

Using our collected marathon results, we kept the men's qualifying times set at the current Boston qualifying times and determined from the data the proportion of men that qualify in each age group (AG), marathon, and year. That is, the proportion of men qualifying is specified from the data and considered fixed. The proportion of men qualifying at time $t$ is calculated as shown in Eq 1. The proportion of women qualifying is found in the same manner.

$$propM_{ijk}(t) = \frac{\#\text{ of Men Qualifying in AG } i, \text{ marathon } j, \text{ year } k, \text{ at qualifying time } t}{\#\text{ of Men in AG } i, \text{ marathon } j, \text{ and year } k} \quad (1)$$

We minimized the difference in the proportion of men and women qualifying across each marathon and year as shown in the objective function given in Eq 2. We performed a grid search at every one minute interval between the age-group-specific Men's QT plus twenty minutes and plus forty minutes, and evaluated the objective function within this range for

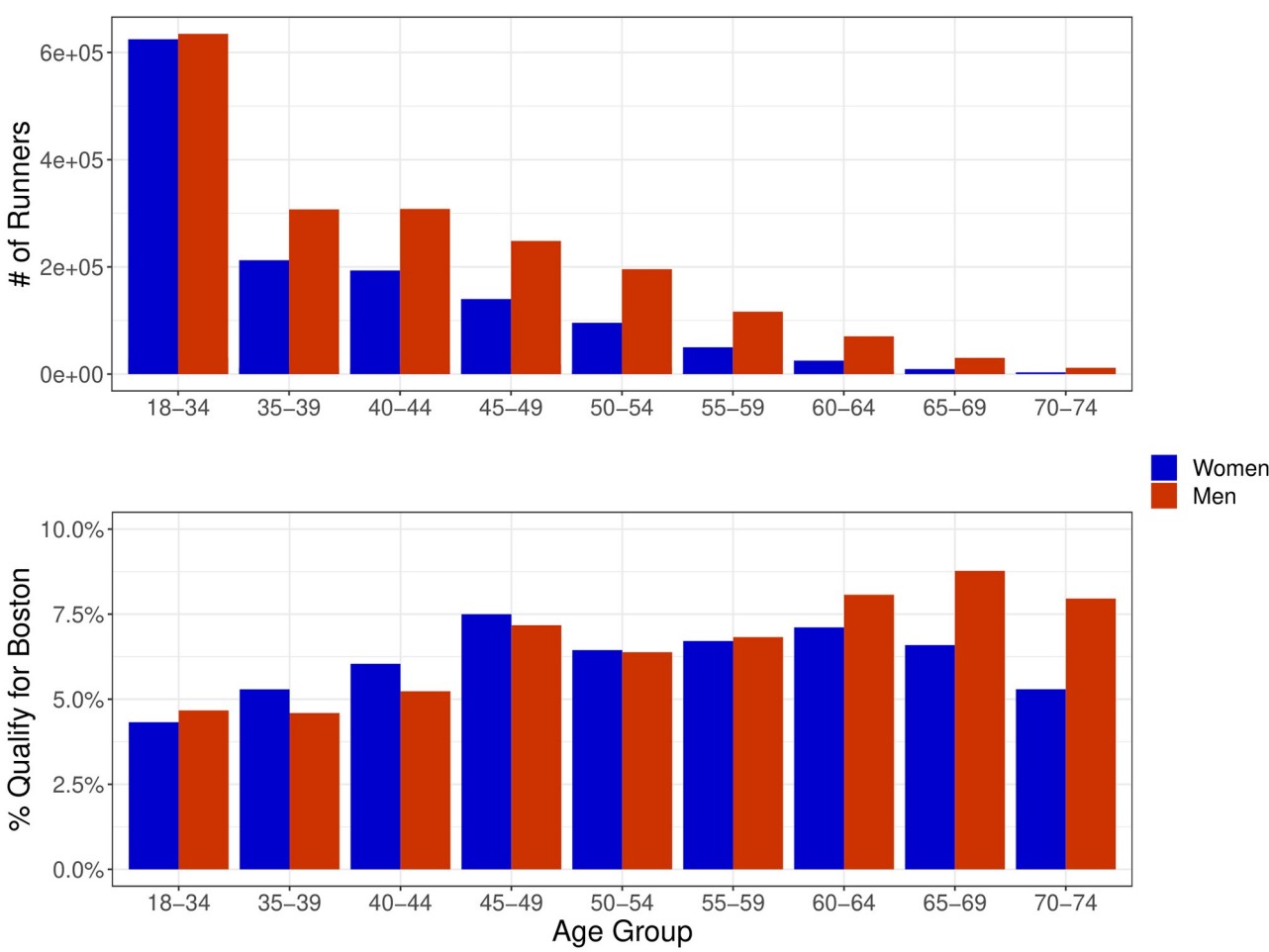

**Fig 1. Number of men and women runners in each age group and percentage that qualify for Boston under the 2020 qualifying times across all marathons collected.**

each age group. For example, for the 18–34 age group with a Men's QT of 3 hours, we evaluated the objective function (Eq 2) with *t* equal to every minute between 3 hours and 20 minutes and 3 hours and 40 minutes. We refer to the value of *t* that minimized this objective function as the optimized difference in proportions qualifying time (Optimized QT). The term 'optimize' in this context specifically refers to the process of determining the women's qualifying times that minimize the absolute differences in proportions of male and female qualifiers, as defined by the chosen objective function in Eq 2. We inspected all plots of the objective function versus qualifying times to ensure a true minimum was reached.

$$\text{Optimized QT} = \min_{t} \sum_{jk} |propW_{ijk}(t) - propM_{ijk}| \qquad (2)$$

Systematic differences in marathons such as cut-off time, elevation, and prize money can affect the field of runners in each marathon. To ensure our results are robust and not sensitive to the choice of the sample of marathons included, we repeated this analysis ten times excluding one marathon at a time and compared the resulting ten qualifying times to the qualifying times found with all marathons included.

### Proportion analysis

We used a two proportion z-test to determine if the differences in proportions between men and women are significant in each marathon and year. We set a threshold on the proportion test to a p-value less than 5%. We performed the proportion test at both the current women's Boston qualifying times and at the times found when minimizing the difference in proportions for comparison.

### Results

The distribution of the differences between men's and women's times for the top 4–9% of runners are shown in the boxplot in Fig 2. The red line indicates the 30-minute time gap currently used in each age group. A difference of less than 30 minutes exists for the 35–39, 40–44, and 45–59 age groups and the time difference is greater than 30 minutes for all of the 60+ age groups.

The qualifying times found from minimizing Eq 2 over *t* are shown in Table 2. When compared to the Current QTs, the Optimized QT for the 18–34 age group was 1 minute slower, the

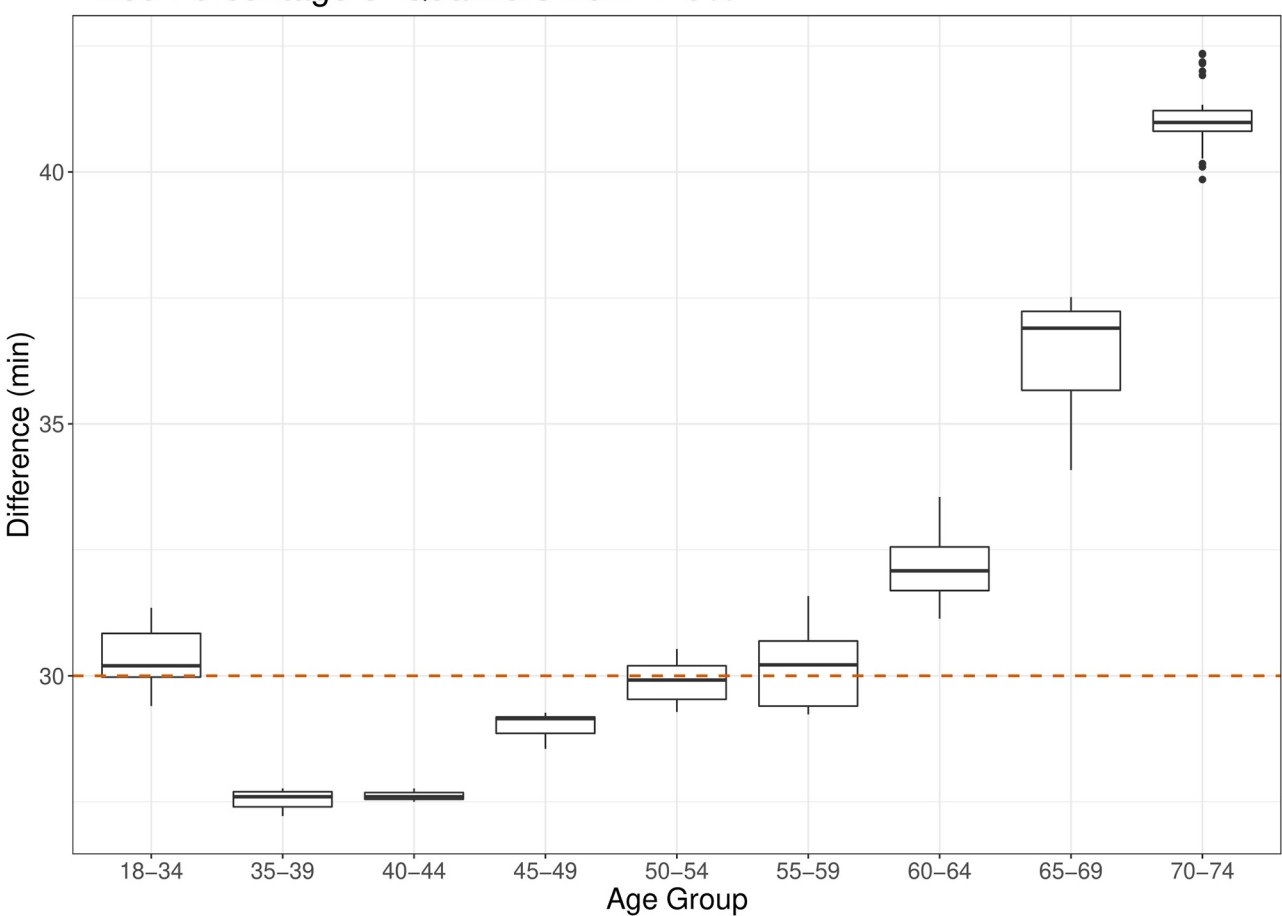

**Fig 2. Difference between men's and women's times for the top 4–9% of runners at each 0.1% increment in each age group.** The red line indicates the 30-minute difference which is currently used in each age group.

**Table 2. Optimized women's qualifying times found from Eq 2.** The difference in minutes between the Current QTs and the Optimized QTs are given in the second row. The absolute difference in the number of qualifiers at the Current QTs compared to the Optimized QTs is shown in the third row with the relative percent change displayed in parentheses. The last row gives the percent difference of the proportion of men and the proportion of women that qualify at their respective qualifying times. A negative value indicates a higher proportion of men qualify relative to women.

| Age Group | 18–34 | 35–39 | 40–44 | 45–49 | 50–54 | 55–59 | 60–64 | 65–69 | 70–74 |
|---|---|---|---|---|---|---|---|---|---|
| Optimized Women's QT | 3:31 | 3:32 | 3:37 | 3:48 | 3:54 | 4:05 | 4:19 | 4:39 | 4:55 |
| Difference from Current QT (minutes) | +1 | -3 | -3 | -2 | -1 | 0 | -1 | +4 | +5 |
| Change in # of Qualifiers at Optimized QT (Relative % Change) | +1579 (5.9%) | -1911 (-17.0%) | -1997 (-17.1%) | -1173 (-11.2%) | -357 (-5.8%) | 0 (0%) | -81 (-4.5%) | +110 (+17.9%) | +38 (+22.8%) |
| % Difference proportion of men vs women qualifying at Optimized QTs (% Difference at Current QTs) | -0.09% (-3.47%) | -0.20% (+0.70%) | -0.23% (+0.81%) | -0.51% (+.33%) | -0.31% (+0.06%) | -0.11% (-0.11%) | -1.28% (-0.95%) | -1.00% (-2.18%) | -1.46% (-2.66%) |

65–69 age groups was 4 minutes slower, and the 70–74 age group was 5 minutes slower. All other age groups were between 0 and 3 minutes faster than the Current qualifying times. Using the Optimized QT instead of the Current QT has a large effect on the absolute and relative change of the number of women qualifying in some age groups. For example, a decrease of 3 minutes for the 35–39 and 40–44 age groups would lead to over 1900 fewer qualifiers in each age group (approximately a 17% relative decrease). An increase of 4 minutes at the 65–69 age group and 5 minutes in the 70–74 age groups would increase the number of qualifiers by 110 and 38 runners respectively (17.9% and 22.8% relative increase). In all age groups except 60–64, the difference in proportions of men and women that qualify are closer to zero using the Optimized QTs.

To assess the sensitivity of our approach to the choice of marathons included, we also determined the qualifying times by excluding one marathon at a time and again minimizing the difference in proportions. The frequency at which each qualifying time was obtained is shown in Fig 3. The qualifying times found when minimizing the difference in proportions across all races, as reported in Table 2, is highlighted in red. In most cases, the results when excluding one marathon are all within 1 minute of the Optimized QTs found when all marathons are included. The 70–74 age group was more sensitive to the exclusion of individual marathons with Optimized QTs ranging from 4:51–5:03.

Results of the two proportion z-test are shown in Fig 4. The first column of Fig 4 shows the results of the proportion test evaluated at the current women's qualifying times and the middle column shows the results at the optimized difference in proportions women's qualifying times. Red squares indicate a higher proportion of men qualifiers and blue squares indicate a higher proportion of women qualifiers. To represent significant differences between men and women's proportions, we set a threshold on the proportion test to a p-value less than 5%. Dark red and dark blue squares indicate the difference is significant at a 95% confidence level. Summaries by age group of the number of results at each significance level at the Current QTs and Optimized QTs are shown in the third column of Fig 4.

## Discussion

The Boston Marathon currently uses a static 30-minute time gap between men's and women's qualifying times across each age group, implying women become faster relative to men as they age. We collected data to evaluate the impact of this 30-minute time gap on qualifiers across different age groups and proposed a data-driven approach to find qualifying times that maintain parity within each age group.

We collected data from ten different marathons from 2000–2019. Overall, our data set includes 3.2 million runners. Based on data provided by the BAA, the marathons in our data

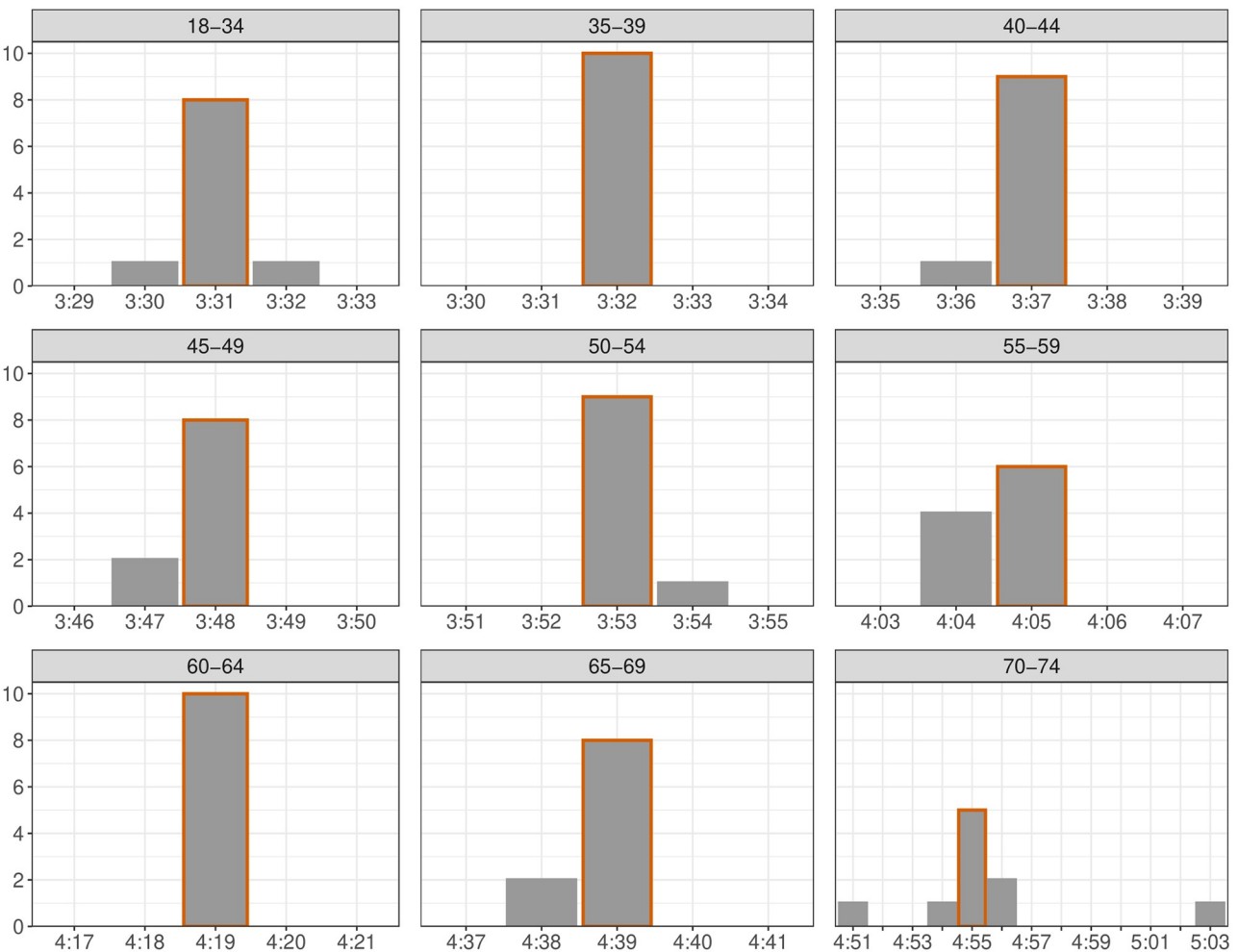

**Fig 3. Optimized difference in proportions QTs found when excluding one of the ten marathons at a time.** The Optimized QTs across all marathons is highlighted in red.

account for 28.6% of all qualifiers that entered the Boston Marathon in 2018 and 2019. The geographic distribution of runners' home states in our data closely resembled that of the Boston Marathon data. We also compared our data with the report by www.runrepeat.com using their database of races which is said to include 96% of all US race results from 1986 to 2018 [14]. We observed that the average finishing times, as well as the age and sex distribution of runners in our data, are consistent with the corresponding values given in their report (S2 and S3 Figs). Thus, our data set appears to be a reasonable sample of marathon race results. For validation purposes, future work could include replicating this analysis on another data set with different marathons.

Based on our data, we find that in the 18–34 age group, the proportion of men who qualify for the Boston Marathon at the current qualifying times is slightly higher than that of women. In the 35–49 age groups, however, more women qualify proportionally than men. In the 50–59 age groups, the proportions are similar between men and women. Finally, in the 60–74 age groups, the proportion of men who qualify is higher, and the gap between the proportion of men and women continues to increase with each age group. As an alternative to a set time gap, we determined qualifying times by minimizing the difference in the proportion of men and

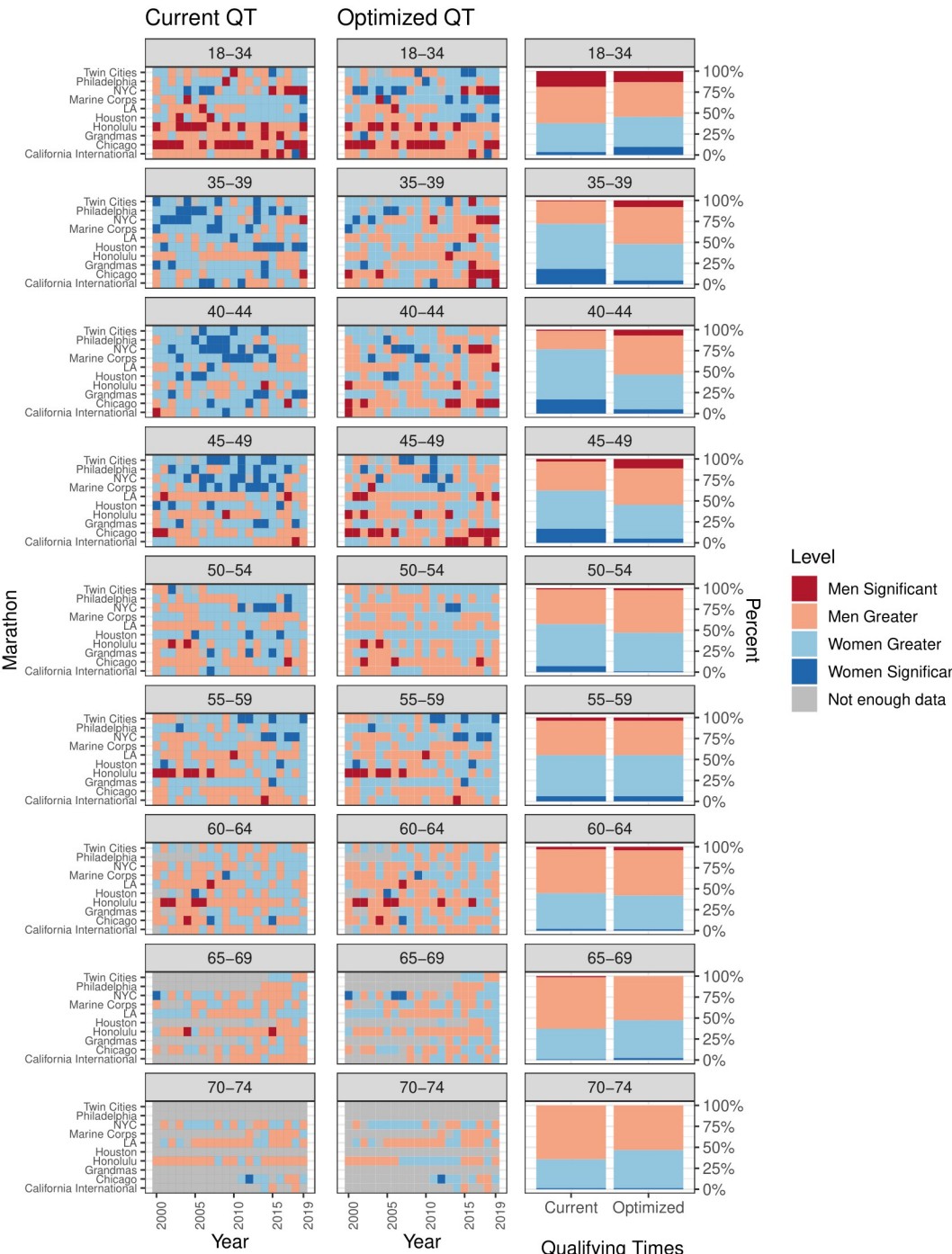

**Fig 4. The first column shows the results of the proportion test using the current Boston marathon qualifying times, the second column shows the results using the optimized difference in proportions qualifying times given in Table 2, and the third column shows the total number of results for each level at the current and optimized difference in proportions qualifying times.** Red squares represent a higher proportion of men qualifying, blue squares represent a higher proportion of women qualifying. Dark red and dark blue indicate whether this difference is significant at a 95% significance level while light red and light blue indicate a difference in proportion that is not significant at the 95% level.

women qualifiers across all marathons and years. Our method suggests adding 1 minute in the 18–34 age group, adding 4–5 minutes in the 65–69 and 70–74, and subtracting 3 minutes in the 35–39 and 40–44 age groups from the respective current women's qualifying times would lead to a more equitable proportion of women to men qualifiers.

In contrast to the Boston Marathon, the New York City Marathon, which utilizes a mixture of entrants through qualifying times, lottery entries, and charity entries, has qualifying times that range from a 20 minute time gap for the 18–34 age group to 100 minutes for the 80+ age group [15]. Previous studies have indicated the decline in running performance for women is greater with age than for men [16, 17]. Fig 2 illustrates a discrepancy in finishing times between men and women in the top 4–9% of runners in our data, suggesting a time difference that increases with age may be more appropriate than a static time gap.

We kept the men's qualifying times fixed and found qualifying times for women to minimize the difference in the proportion of women and men qualifying across each age group, marathon, and year. Many potential objective functions could be used to determine qualifying times in this manner. We chose to keep the proportion of men qualifying constant to ensure our Optimized QTs did not cause large reductions in qualifiers in the older age groups. At the current QTs, the proportion of qualifiers in the older age groups is higher than it is in the younger age groups (Fig 1). Therefore, if we allowed the men's qualifying times to change as well and optimized to find a consistent proportion of qualifiers across all age groups, this would decrease the number of qualifiers in the older age groups and increase the number of qualifiers in the younger age groups. In 2002 the BAA altered the qualifying times to make the times easier for all runners 45 and older to accommodate more participants [18]. We infer this decision was motivated by a desire to boost participation among older runners specifically.

We minimized the sum of the absolute differences in proportions as it is robust to outliers and ensures an equal proportion of qualifiers across genders in each age group. However, this approach resulted in fewer women qualifying in some of the younger age groups than under the previous time standards. To address this issue, future research could consider adding constraints to hold the number of female qualifiers fixed. However, such constraints could have unintended consequences, such as favoring the current qualifying times, particularly in the younger age groups. Therefore, any future work should carefully consider these trade-offs when attempting to mitigate the issue of having fewer women qualifiers.

The Optimized QTs still result in faster times for women relative to men as they age but the differences are slightly less extreme than at the Current QTs. This could be explained by the lower popularity of recreational marathon running with older women than men. Additionally, cut-off times used in many races could disproportionately discourage women in older age groups from participating. In general, it has been observed that more older men than women participate in sports activities [19]. Women were not even allowed to participate in marathons fifty years ago [20]. However, the relative participation of women in marathon running has been increasing in recent years and on average, women are improving faster relative to men [14, 17, 21]. We also observe this in our data set over the past 20 years (S3 Fig). As running continues to become more popular, this dynamic may change over time.

The BAA utilizes a rolling enrollment system which allows the faster participants relative to their qualifying times to register first. In all but one year since the rolling system was implemented in 2012, not all runners who qualified were accepted into the race [18]. The effect of these faster cut-off times is outside the scope of our analysis. The number of qualifiers does appear to be sensitive to small changes in qualifying times, which could have a significant effect on the proportion of qualifiers and would be useful to include in a future study.

The Boston Marathon was canceled for the first time in history in 2020 due to the COVID-19 pandemic [22]. We only collected data and explored marathons for the 20 years prior to the

pandemic but hypothesize it may take awhile for road racing to rebound to pre-pandemic popularity. For the first time since 2013, all runners who qualified and entered Boston were accepted to run it in 2022 and 2023, despite the fact the field size remained unchanged compared to 2015–2019 [18]. The abundance of cancelled or postponed races in 2020 and 2021, which limited the opportunities for runners to achieve a qualifying time, may have contributed to the easier entrance times. The effect of the pandemic on road race participation across genders and age groups will be important to consider to compare future data against our results. A dynamic perspective could be applied to monitor changes in the proportion of qualifiers over time, particularly in response to external events like the pandemic, and to examine the differential impacts across genders and age groups.

Qualifying for Boston is the goal of many amateur runners who may run multiple marathons in an effort to achieve this goal. Runners training to qualify for Boston are likely to have finishing times clustered around their respective qualifying times. Thus, small changes have a large impact on the number and proportion of women that qualify (Table 2). Due to the popularity of the race, the BAA has repeatedly altered the qualifying times to limit the size of the field. We utilized a data-driven approach to setting new qualifying times that maintain parity within each age group. Overall, the qualifying times found using our method and data set are similar to the current qualifying times suggesting the static 30-minute time gap is consistent with our data-driven approach. However, we find modest changes to the women's qualifying times would lead to a more equal proportion of qualifiers across genders. Our method could be easily adapted to other potential objective functions to determine qualifying times specific to the race organizer's goals and desired distribution of qualifiers.

## Conclusion

In our data set, we observe the proportion of men that qualify at the current Boston qualifying times is slightly higher than women in the 18–34 age group in our data, in the 35–49 age groups more women qualify proportionally than men, in the 50–59 age groups the proportions are similar, and at the 60–74 age groups the proportion of men qualifying is higher and the gap between the proportion or men and women grows in each age group. As an alternative to the 30-minute fixed time gap currently used, we determined qualifying times by minimizing the difference in the proportion of men and women qualifiers across all marathons and years. Our analysis shows adjusting the qualifying times by adding 1 minute to the 18–34 age group, adding 4–5 minutes in the 65–69 and 70–74 age groups, and subtracting 3 minutes from the 35–39 and 40–44 age groups from the current women's qualifying times could result in a more equal proportion of women to men qualifiers. Our methodology could easily be applied to other potential objective functions to determine data-informed qualifying times specific to the race organizer's desired size and distribution of qualifiers.

## Supporting information

**S1 Fig. Distribution of finishing times by marathon.** The total number of runners observed in each marathon is shown in parentheses. The orange vertical line indicates the cut-off time for each marathon. Some marathons are strict with their cut-off times while others are more lenient. An interesting feature to note, in each marathon except for Honolulu we see a bump right around the 4 hour mark, a popular goal time for many runners.
(TIF)

**S2 Fig. Distribution of all finishing times by age group and gender.** The number of runners in each age group and gender are shown in parentheses. The distribution of times does not

change much between the 18–34, 35–39, and 40–44 age groups. We then see a shift towards slower times in each subsequent age group, particularly in the oldest age groups.
(TIF)

**S3 Fig. Percentage of women runners by age group from 2000–2019.** The 18–34 age group increased towards 50% and then appears to have leveled off around that 50% split. All other age groups are increasing on average each year. We observe a sharper increase in the percentage of Women in many age groups in 2012 due to New York not being held that year. New York has a lower percentage of women runners than most other marathons so it's exclusion leads to the spike observed.
(TIF)

**S1 Table. Number of participants by marathon and age group.**
(PDF)

**S2 Table. Cut-off times and average number of participants by marathon.** Differences in cut-off times may be responsible for the variation in the proportion of qualifiers across different marathons. Also, each marathon has different levels of enforcement with cut-off times which could impact the distribution of finishers.
(PDF)

## Author Contributions

**Conceptualization:** Laura Albrecht, Ross Ring-Jarvi, Dorit Hammerling.

**Data curation:** Laura Albrecht, Ross Ring-Jarvi.

**Formal analysis:** Laura Albrecht.

**Methodology:** Laura Albrecht, Dorit Hammerling.

**Supervision:** Dorit Hammerling.

**Visualization:** Laura Albrecht.

**Writing – original draft:** Laura Albrecht, Ross Ring-Jarvi.

**Writing – review & editing:** Dorit Hammerling.

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
