## [Decision Letter · Decision Letter 0]

23 Jun 2022

PONE-D-22-13227Evaluating the fairness of the Boston Marathon qualifying timesPLOS ONE

Dear Dr. Albrecht,

Thank you for submitting your manuscript to PLOS ONE. After careful consideration, we feel that it has merit but does not fully meet PLOS ONE’s publication criteria as it currently stands. Therefore, we invite you to submit a revised version of the manuscript that addresses the points raised during the review process.

In order to resolve the differences between reviewers, I have read the paper and recommend significant revision. Please review each reviewers comments and address their concerns. In addition, I recommend the following:I believe reviewer 2 makes a legitimate case for the use of fairness. While I believe you have defined the word, I question whether fairness is relevant in a scientific paper. I believe you should reposition your paper as evidence-based for qualifying times. This is a more direct way to direct your argument; *Are the current standards evidence-based? It doesn't appear so. Is your approach better? I can't say, but we should establish standards based on the evidence, not arbitrary fixed numbers. *That is the case the authors should make and a suggestion on how to get there!  There is inconsistent grammar, language use (like many in areas where present not past tense is used). The paper feels disjointed as if written by different individuals and then combined without regard to flow.There are insufficient references for a paper of this magnitude. In particular, much of the (limited) discussion is more extended conclusions and speculation and lacks any meaningful literature to support it. I recommend several references to review:Bagley DOI: 10.1111/sms.13404Tanaka DOI: 10.1113/jphysiol.2007.141879Nicolaidis doi: 10.3390/ijerph16101777Waldvogel doi: 10.3390/ijerph16132377https://runrepeat.com/state-of-runninghttps://runnerclick.com/marathon-finishing-times-study-and-statistics/The introduction is extensive and well written but should end with a clear purpose and hypotheses. The Methods are very confusing and poorly structured, and should be written in the past tense. The general writing is completely different from the intro and much of the methods are actually results. I recommend streamlining and reorganizing to clearly identify who the participants were, where the data came from, how it was analyzed with details on stats, and specific criteria for exclusion of data. I envision much of the data with figures should move to results.Results are extensive but should be streamlined and flow based on the story you with to tell in the discussion. Figures are at times confusing with superfluous text on the figures. Titles need to be concise where possible and clearly identify what the figure is showing without excess story or text. I'm just having a hard time following the story you're telling.**I recommend identifying the key 3 or 4 figures and moving the rest to supplements to tell your story. **The discussion is inadequate for the paper and data you present. Two big paragraphs do little to help the paper and largely just try to tell us how much better your optimized times are. However, you haven't really made that case, and really should leave it to the conclusion and future rec's for research. It also needs to lean into supporting references. I suggest:Restating the purpose and hypotheses then reviewing what you found. Then each subsequent paragraph targets a single finding.An additional paragraph can cover unexpected findings and then perhaps one more paragraph to note the limits of this study and future recommendations.Now hit the conclusion with the overarching findings and message. Avoid over speaking your results.Please utilize the time available and perhaps request an independent reader prior to resubmission. We look forward to reading your revised work.

We look forward to receiving your revised manuscript.

Kind regards,

Chris Harnish, PhD

Academic Editor

PLOS ONE

Journal Requirements:

2. In your Methods section, please include additional information about your dataset and ensure that you have included a statement specifying whether the collection and analysis method complied with the terms and conditions for the source of the data.

Reviewers' comments:

Reviewer's Responses to Questions

**Comments to the Author**

1. Is the manuscript technically sound, and do the data support the conclusions?

Reviewer #1: Yes

Reviewer #2: No

2. Has the statistical analysis been performed appropriately and rigorously? 

Reviewer #1: Yes

Reviewer #2: Yes

3. Have the authors made all data underlying the findings in their manuscript fully available?

Reviewer #1: Yes

Reviewer #2: Yes

4. Is the manuscript presented in an intelligible fashion and written in standard English?

Reviewer #1: Yes

Reviewer #2: No

5. Review Comments to the Author

Reviewer #1: This study presents the results of primary scientific research. To this reviewer’s knowledge, the results reported in this manuscript have not been published elsewhere. Statistics and other analysis are performed to a high technical standard and are described in sufficient detail. Conclusions are presented in an appropriate fashion and are supported by the data. This article is presented in an intelligible fashion and is written in standard English. In addition, the researchers meet all applicable standards for the ethics of experimentation and research integrity.

I am curious on why those ten races were selected for data analysis. It is not noted in the methods. When looking at the top qualify races for Boston, several of them are not even listed (e.g. Honolulu, Houston, LA). Top Qualifying Races | Boston Athletic Association (baa.org). Please provide more information on this.

This reviewer suggests providing a shorter summary after each Figure. The current summaries are very lengthy and are what is exactly in the text of the article. In addition, the color coding is helpful but many readers will be looking at this article in black/white (especially if print). This reviewer would suggest looking at different ways to express the data so can be viewed by all. Since, this study looked at data prior to the pandemic, do the authors feel that this has now changed the outcome of their data. Boston has now opened up the race for more athletes to be able to run. This might be something to mention in the discussion. Overall, I enjoyed reading this article. As a runner myself this is always great to see how investigators are looking at marathon data.

Reviewer #2: Dear authors,

The study aimed to investigate whether the Boston Marathon qualifying times are “Fair”. Although this is an interesting topic and deserves attention, in a way that future marathons' qualifying times seek to include a more homogenous number of qualifiers across genders, I have some concerns about the way the manuscript was presented and reported.

First, I believe that if authors want to evaluate fairness, they should include a proper definition of the term and how this affected their research question. Fairness in sports is a rather complicated matter, involving several other factors that were not considered in the study and not discussed properly. In my opinion, the study simply aimed to analyse whether the current Boston qualifying times include a similar percentage of women and men, and whether the times could be changed by applying a mathematical model in a way that the difference in proportion of qualifiers across genders is smaller (although the authors do not report how much this difference is).

Second, it is not clear from the introduction and the aims of the study what is the current differences in the number of women and men that qualify in each race. This is a piece of very important information, that would allow us to know if there is indeed a large difference in the proportion of qualifiers across genders.

Also, the methods section is quite complicated to follow and there is some information that is not relevant to the aims of the study. For example, the authors surprisingly collected data from 10 different other marathons, and it is not clear why they did that. I believe most of the data that were analysed and presented are not relevant to the aims of the study which was to analyse whether the Boston qualifying times include a similar proportion of runners across genders. I believe the authors should then focus on the “Proportion Analysis” assessing the Boston qualifying times and how changing it would minimize the difference in the proportion of qualifiers across each race. The authors also included in the methods sections, several results from their analysis that should be included only in the “Results” section.

Finally, the “Discussion” section does not represent all the data that was analysed. The authors previously presented the results from several different analyses, yet the discussion had only two paragraphs and only included one reference. The authors should aim to discuss the results of their study comparing it with previous published studies, making sure that all the data that was analysed, is discussed as well.

Kind regards.

6. PLOS authors have the option to publish the peer review history of their article (what does this mean?). If published, this will include your full peer review and any attached files.

Reviewer #1: No

Reviewer #2: No

---

## [Author Response · Author response to Decision Letter 0]

3 Aug 2022

Editor #1, comment #1

I believe reviewer 2 makes a legitimate case for the use of fairness. While I believe you have defined

the word, I question whether fairness is relevant in a scientific paper. I believe you should reposition your

paper as evidence-based for qualifying times. This is a more direct way to direct your argument; Are the

current standards evidence-based? It doesn’t appear so. Is your approach better? I can’t say, but we should

establish standards based on the evidence, not arbitrary fixed numbers. That is the case the authors should

make and a suggestion on how to get there!

Our response #0.1

We think this is an excellent suggestion. We agree presenting our method as an evidence-based alternative

is clearer and less subjective. We have repositioned our paper as the editor suggested and have removed the

discussion of the word fairness.

Editor #1, comment #2

There is inconsistent grammar, language use (like many in areas where present not past tense is used).

The paper feels disjointed as if written by different individuals and then combined without regard to flow.

Our response #0.2

We have edited the paper to make sure the tense is consistent throughout each section and have added

some transitions to try and make the paper flow better.

Editor #1, comment #3

There are insufficient references for a paper of this magnitude. In particular, much of the (limited)

discussion is more extended conclusions and speculation and lacks any meaningful literature to support

it. I recommend several references to review: Bagley DOI: 10.1111/sms.13404 Tanaka DOI: 10.1113/jphysiol.

2007.141879 Nicolaidis doi: 10.3390/ijerph16101777Waldvogel doi: 10.3390/ijerph16132377 https://runrepeat.com/stateof-

running https://runnerclick.com/marathon-finishing-times-study-and-statistics/

Our response #0.3

Thank you so much to the editor for taking the time to include these references as a starting point. We

have expanded the discussion to incorporate more references, including several of those suggestions, that are

relevant to our hypothesis and results.

Editor #1, comment #4

The introduction is extensive and well written but should end with a clear purpose and hypotheses.

Our response #0.4

Thank you. An additional paragraph has been added to the end of the introduction to make our purpose

more clear and help ease the transition into the methods section.

Editor #1, comment #5

The Methods are very confusing and poorly structured, and should be written in the past tense. The

general writing is completely different from the intro and much of the methods are actually results. I

recommend streamlining and reorganizing to clearly identify who the participants were, where the data

came from, how it was analyzed with details on stats, and specific criteria for exclusion of data. I envision

much of the data with figures should move to results.

Our response #0.5

Thank you for bringing this to our attention. We have revised the methods section to try and make it

more clear and hope the tone fits better with the rest of the paper now. Most of the figures and exploratory

data analysis have been moved to either the results section or the supplemental information.

Editor #1, comment #6

Results are extensive but should be streamlined and flow based on the story you with to tell in the

discussion.

Our response #0.6

We have updated the results section to to better focus our story. Fig 6 has been moved to the supplemental

information section.

Editor #1, comment #7

Figures are at times confusing with superfluous text on the figures. Titles need to be concise where

possible and clearly identify what the figure is showing without excess story or text. I’m just having a hard

time following the story you’re telling. I recommend identifying the key 3 or 4 figures and moving the rest

to supplements to tell your story.

Our response #0.7

We have updated all the figure captions to be more concise. We really appreciate the suggestion to

identify the key figures. This really helped us focus our story and improved the flow of our paper. We

have moved four of the figures into the supplemental section. While we believe these figures give interesting

insight into our data set, we agree the supplemental section is probably a better place to include them.

Editor #1, comment #8

The discussion is inadequate for the paper and data you present. Two big paragraphs do little to help

the paper and largely just try to tell us how much better your optimized times are. However, you haven’t

really made that case, and really should leave it to the conclusion and future rec’s for research. It also

needs to lean into supporting references. I suggest: Restating the purpose and hypotheses then reviewing

what you found. Then each subsequent paragraph targets a single finding. An additional paragraph can

cover unexpected findings and then perhaps one more paragraph to note the limits of this study and future

recommendations.

Our response #0.8

Thank you for this very specific outline on how to improve our discussion. We have thoroughly reorganized

the discussion to follow this outline. We have expanded our discussion to include more references and

added a discussion about the limitation of our work and potential directions of future work.

Editor #1, comment #9

Now hit the conclusion with the overarching findings and message. Avoid over speaking your results.

Our response #0.9

We have revised the conclusion in this way. We truly thank you for taking the time to provide such

detailed and useful suggestions for each section of our paper.

Reviewer #1, comment #1

I am curious on why those ten races were selected for data analysis. It is not noted in the methods. When

looking at the top qualify races for Boston, several of them are not even listed (e.g. Honolulu, Houston, LA).

Top Qualifying Races — Boston Athletic Association (baa.org). Please provide more information on this.

Our response #1.1

Thank you for bringing it to our attention that we failed to explain our choice of marathons. We chose

these 10 marathons to get a good geographic sample of marathons that are 10 of the top 15 largest marathons

in the US. Additionally, all marathons except for Honolulu are listed as top qualifying races for Boston. We

have added an explanation of this in the Data section.

Reviewer #1, comment #2

This reviewer suggests providing a shorter summary after each Figure. The current summaries are very

lengthy and are what is exactly in the text of the article. In addition, the color coding is helpful but many

readers will be looking at this article in black/white (especially if print). This reviewer would suggest looking

at different ways to express the data so can be viewed by all.

Our response #1.2

Thank you for the suggestion. The figures summaries have been shortened on all figures. The color

choices in our figures all come from color blind friendly color palettes and as this is an online only journal,

we believe they should be readable to readers.

Reviewer #1, comment #3

Since, this study looked at data prior to the pandemic, do the authors feel that this has now changed

the outcome of their data. Boston has now opened up the race for more athletes to be able to run. This

might be something to mention in the discussion.

Our response #1.3

This is an interesting point we hadn’t considered yet. We have added a paragraph in the discussion

regarding the effect of the pandemic on marathons and how this will impact future data. We only collected

data on and explored marathons prior to the pandemic but hypothesize it may take a few years for road

racing to rebound to pre-pandemic popularity. For the first time since 2013, all runners who qualified and

entered Boston were accepted to run it this year. The field size for the 2022 race was the same as the previous

5 years before the pandemic. We assume the abundance of canceled or postponed races in 2020 and 2021,

which limited the opportunities to achieve a qualifying time, may have been a big contributing factor to the

easier entrance criteria this year. The effect of the pandemic on road race participation across genders and

age groups will be interesting to look at in the coming years.

Reviewer #2, comment #1

First, I believe that if authors want to evaluate fairness, they should include a proper definition of

the term and how this affected their research question. Fairness in sports is a rather complicated matter,

involving several other factors that were not considered in the study and not discussed properly. In my

opinion, the study simply aimed to analyse whether the current Boston qualifying times include a similar

percentage of women and men, and whether the times could be changed by applying a mathematical model

in a way that the difference in proportion of qualifiers across genders is smaller (although the authors do not

report how much this difference is).

Our response #2.1

We agree that the discussion of fairness is complicated and have instead reframed our paper, as suggested

by the editor, away from fairness. In Fig 4 we show a side by side bar plot of the results of the proportion test

at the current Boston qualifying times and our optimized times to try and show that our optimized times

lead to a more equal proportion qualifiers. We have edited the explanation of this plot to try and highlight

that result more explicitly. We have also added a row to Table 2 showing the difference in proportions

between men and women across all races at the current and optimized times.

Reviewer #2, comment #2

Second, it is not clear from the introduction and the aims of the study what is the current differences in

the number of women and men that qualify in each race. This is a piece of very important information, that

would allow us to know if there is indeed a large difference in the proportion of qualifiers across genders.

Our response #2.2

Thank you and we agree we skimmed over detail and it needs to be more clear. In the second row of

Fig 1 we show the proportion of men and women that qualify in our data in each age group. Other than

the 18-34 age group which has a slightly higher percentage of men qualifying than women, women qualify at

a higher rate for all ages 50 and under. For age groups 55 and over, the percentage of men that qualify is

greater and the gap between the proportion of men and women qualifying increases in each subsequent age

group. This explanation has been added to the Data section.

Reviewer #2, comment #3

Also, the methods section is quite complicated to follow and there is some information that is not

relevant to the aims of the study. For example, the authors surprisingly collected data from 10 different

other marathons, and it is not clear why they did that. I believe most of the data that were analysed and

presented are not relevant to the aims of the study which was to analyse whether the Boston qualifying

times include a similar proportion of runners across genders. I believe the authors should then focus on

the “Proportion Analysis” assessing the Boston qualifying times and how changing it would minimize the

difference in the proportion of qualifiers across each race. The authors also included in the methods sections,

several results from their analysis that should be included only in the “Results” section.

Our response #2.3

Thank you for this feedback. We have edited the methods section to hopefully make it easier to follow.

We have also moved any results in the methods section into the results section. As described above, we have

added an explanation to why these 10 marathons were chosen. Many of the exploratory data pieces were

moved to the supplemental information as we think it may be of interest to reader, especially those in the

running community, but agree it detracts from the story we are trying to tell.

Reviewer #2, comment #4

Finally, the “Discussion” section does not represent all the data that was analysed. The authors previously

presented the results from several different analyses, yet the discussion had only two paragraphs and

only included one reference. The authors should aim to discuss the results of their study comparing it with

previous published studies, making sure that all the data that was analysed, is discussed as well.

Our response #2.4

Thank you for this suggestion. We have expanded the discussion section to ensure all of the results presented are included. The discussion has been expanded as described in response to the editor’s comments

to include a more thorough comparison with previous literature.

---

## [Decision Letter · Decision Letter 1]

23 Aug 2022

PONE-D-22-13227R1Data-driven evaluation of the Boston marathon qualifying timesPLOS ONE

Dear Dr. Albrecht,

Thank you for submitting your revised manuscript to PLOS ONE. While I believe the manuscript has merit, reviewers could come to a consensus on this. Therefore, after careful review I enlisted a third non-academic reviewer with significant expertise in statistics and data analysis. It has been enlightening to read a fresh review from a unique point of view. I believe reviewer 3 has offered the authors a clear pathway forward while improving the overall value of the manuscript. Therefore, we invite you to submit a revised version of the manuscript that addresses the key areas recommended below.Please give careful consideration to the statistical methods provided.Consider carefully reviewing several databases on running, like that of run repeat. Addressing the Fairness question; *reviewers struggled with this concept and I would decide on one clear path and stay with it. Reviewer 3 offers some ideas.*Be clear on what your messaging is on the results. *Editor note: I believe you've overplayed the grand outcomes and missed some key points, like the Boston qualifier really breaks down as a specific age. **I believe reviewer 3 has offered some of the best advice to tackle the results issues, but it is a bit of work**.*I appreciate your understanding and I look forward to reading the next revision. Please submit your revised manuscript by Oct 07 2022 11:59PM. If you will need more time than this to complete your revisions, please reply to this message or contact the journal office at plosone@plos.org. Please include the following items when submitting your revised manuscript:A rebuttal letter that responds to each point raised by the academic editor and reviewer(s). You should upload this letter as a separate file labeled 'Response to Reviewers'.A marked-up copy of your manuscript that highlights changes made to the original version. You should upload this as a separate file labeled 'Revised Manuscript with Track Changes'.An unmarked version of your revised paper without tracked changes. You should upload this as a separate file labeled 'Manuscript'.

We look forward to receiving your revised manuscript.

Kind regards,

Chris Harnish, PhD

Academic Editor

PLOS ONE

Reviewers' comments:

Reviewer's Responses to Questions

**Comments to the Author**

1. If the authors have adequately addressed your comments raised in a previous round of review and you feel that this manuscript is now acceptable for publication, you may indicate that here to bypass the “Comments to the Author” section, enter your conflict of interest statement in the “Confidential to Editor” section, and submit your "Accept" recommendation.

Reviewer #2: (No Response)

Reviewer #3: (No Response)

2. Is the manuscript technically sound, and do the data support the conclusions?

Reviewer #2: Partly

Reviewer #3: Partly

3. Has the statistical analysis been performed appropriately and rigorously? 

Reviewer #2: Yes

Reviewer #3: No

4. Have the authors made all data underlying the findings in their manuscript fully available?

Reviewer #2: Yes

Reviewer #3: Yes

5. Is the manuscript presented in an intelligible fashion and written in standard English?

Reviewer #2: Yes

Reviewer #3: Yes

6. Review Comments to the Author

Reviewer #2: Dear authors,

Thank you for considering my previous comments and making the necessary adjustments. Although I appreciate the authors’ efforts to address all points, there are still some important key points that need to be clarified and properly addressed.

I believe the manuscript is still a bit hard to follow and the message is still not clear. The aim of the study was to analyse how adjusting (not sure if “optimising” is adequate) the qualifying times of the Boston marathon would result in a more equal proportion of men and women across age groups. Yet, there is a lot of information that is not relevant to the aims of the study. For example, I believe most of the figures included in the supplemental material are irrelevant and should be excluded from the study. The introduction has improved and is properly written, clearly stating the problem and how you aim to solve it. However, the other sections are still quite confusing. In the methods section, I had a difficult time trying to understand how the demographics from the other 10 marathons would help you achieve your aim. The study adopted a pretty robust and complicated analysis, so every step of the data collection and analysis needs to be clearly explained. It is also important to report how many runners took part in each race you analysed and also split by age groups.

I still think there are too many figures that do little to help you deliver your message. The authors also did not attach figure 1 to their submission. In your results, you found that decreasing the qualifying times for the 35-39, 40-44, 45-49, 50-54 and 60-64 age groups would “optimise” the distribution of men and women across age groups. However, this approach, at the same time, substantially lowers the overall absolute number of qualifiers in the race, which will, in fact, exclude more runners. For this reason, I don’t think your approach will “optimise” the qualifying times, as you would essentially exclude more people from taking part in the race.

The discussion should start by stating the aims of the study and providing a comprehensive summary of the main findings. The following paragraphs should then expand the main findings comparing your results with previous studies. I think the discussion is quite speculative and not supported by adequate evidence. For example, reference number 17, although interesting, is a blog post that probably was not peer-reviewed. There are also a couple of paragraphs (lines 199 to 214) including information not relevant to the study.

Reviewer #3: This is an interesting piece of work that I think should be published with revisions. Please see attachment: the 'no' under #3 is because 'partly' was not option.

7. PLOS authors have the option to publish the peer review history of their article (what does this mean?). If published, this will include your full peer review and any attached files.

Reviewer #2: No

Reviewer #3: **Yes: **Gregory P. Swinand

---

## [Author Response · Author response to Decision Letter 1]

24 Feb 2023

We address the reviewers comments point by point in the attached document. Thank you for your feedback and suggestions to improve our manuscript.

---

## [Decision Letter · Decision Letter 2]

20 Mar 2023

Data-driven evaluation of the Boston marathon qualifying times

PONE-D-22-13227R2

Dear Dr. Albrecht,

We’re pleased to inform you that your manuscript has been judged scientifically suitable for publication and will be formally accepted for publication once it meets all outstanding technical requirements. In addition, I ask that you address one minor suggestion by our reviewer to highlight the specific impacts on qualifying among women. This can be particularly helpful for those researchers who may wish to build on your research.

Kind regards,

Chris Harnish, PhD

Academic Editor

PLOS ONE

Additional Editor Comments (optional):

Reviewers' comments:

Reviewer's Responses to Questions

**Comments to the Author**

1. If the authors have adequately addressed your comments raised in a previous round of review and you feel that this manuscript is now acceptable for publication, you may indicate that here to bypass the “Comments to the Author” section, enter your conflict of interest statement in the “Confidential to Editor” section, and submit your "Accept" recommendation.

Reviewer #3: All comments have been addressed

2. Is the manuscript technically sound, and do the data support the conclusions?

Reviewer #3: Yes

3. Has the statistical analysis been performed appropriately and rigorously? 

Reviewer #3: Yes

4. Have the authors made all data underlying the findings in their manuscript fully available?

Reviewer #3: Yes

5. Is the manuscript presented in an intelligible fashion and written in standard English?

Reviewer #3: Yes

6. Review Comments to the Author

Reviewer #3: my main concerns were over the sample selection and how they dealt with that. The comparing to the larger database on what i presume are the only available covariates -- age, sex, and time, seems reasonable. the other concern was toning down the idea that this procedure was better or whatever and it is clearly explained that optimal is only with regards to minimising the objective function; which is fine. there is still a curious issue that the proposed new way would lead to fewer women qualifying for certain age categories. for future research, if they want to include, the final note that qualifying has become less difficult in the immediate aftermath of the pandemic is quite interesting, and how these things evolve over time or react to other such world-impacts; maybe a dynamic or prescriptive perspective could be considered next!

7. PLOS authors have the option to publish the peer review history of their article (what does this mean?). If published, this will include your full peer review and any attached files.

Reviewer #3: **Yes: **Dr Gregory P. Swinand

---

## [Editor Report · Acceptance letter]

29 Mar 2023

PONE-D-22-13227R2 

Data-driven evaluation of the Boston marathon qualifying times 

Dear Dr. Albrecht:

I'm pleased to inform you that your manuscript has been deemed suitable for publication in PLOS ONE. Congratulations! Your manuscript is now with our production department. 

Kind regards, 

on behalf of

Dr. Chris Harnish 

Academic Editor

PLOS ONE